# Assessment of Sugar-Related Dietary Patterns to Personality Traits and Cognitive–Behavioural and Emotional Functioning in Working-Age Women

Agnieszka Garbacz [ID], Bogusław Stelcer [ID], Michalina Wielgosik and Magdalena Czlapka-Matyasik *[ID]

Department of Human Nutrition and Dietetics, Poznan University of Life Sciences, Wojska Polskiego 31, 60-624 Poznan, Poland
* Correspondence: magdalena.matyasik@up.poznan.pl

**Abstract:** This cross-sectional study investigated interactions among sugar-related dietary patterns (DPs), personality traits, and cognitive–behavioural and emotional functioning. The study involved working-age women aged 18–54. Data were collected between Winter and Spring of 2020/21. The survey was conducted using anonymised questionnaires. The ten-item personality inventory (TIPI-PL) was used to examine personality traits based on the Big Five personality trait model. A three-factor eating questionnaire (TFEQ-13) was used to measure the following eating behaviours: cognitive restraint (CR), uncontrolled eating (UE), and emotional eating (EE). The KomPAN questionnaire collected the frequency of the intake. Dietary patterns (DPs) were derived by principal component analysis (PCA). A logistic regression (OR) was applied to verify the associations among the DPs, personality traits, and cognitive–behavioural and emotional functioning. Three DPs were identified: sweet-Western (SWDP), pro-healthy (PHDP), and dairy (DDP). Women with high conscientiousness were less likely, by 33%, to adhere to the upper tercile of the SWDP and 80% more likely to the upper tercile of the PHDP. Elevated CR intensity increased by almost twofold (OR: 1.93; $p < 0.001$) the likelihood of high adherence to the SWDP. The high intensity in the EE decreased by 37% (OR: 0.63; $p < 0.01$) the likelihood of increased adherence to the SWDP. Personality traits and eating behaviours significantly correlated with the extracted SWDP.

**Keywords:** sweet taste preferences; diet; personality traits





## 1. Introduction

Statistics report that average sugar consumption has increased worldwide, from 20.71 to 22 kg per inhabitant per year in the last decade [1]. This trend is also continuing in Poland. Since 2010, annual sugar consumption has increased from 39.9 kg to nearly 42 kg per capita [2]. Data indicate that the WHO's recommendations of a maximum of 10% of energy from simple sugars in the diet have been exceeded [3]. Sugar has begun to be seen as an ingredient responsible for the global epidemic of obesity, cardiometabolic diseases, and cancers [4–9]. Therefore, sugar-related dietary behaviours and sweet taste preferences have started to be widely discussed in the literature [7].

Although the influence of sugars on the development of the diseases mentioned above is apparent, the question of what behavioural mechanism leads to the choice of the sweet taste as the dominant taste is still a matter of debate. Sugar-related dietary patterns and their direct translation into personality traits have yet to be defined. We do not know if subjects who consume more sugar tend to reduce their consumption of other food groups or how they control their dietary patterns or express their emotions via dietary choices. Our previous manuscript showed that sugar intake might be related to weekdays and weekend days, and young women, in particular, tend to modify it, reflecting the nutritional value of a daily diet [10]. There needs to be papers comprehensively describing the dietary patterns of individuals in the context of sugar consumption and other eating behaviours, including

fruits, vegetables, dairy, or meat-originating foods. Dietary behaviour has increasingly been attributed to personality types and other psychological factors [11–13]. Psychodietetics is gaining more popularity. However, despite the growing interest in the relationship between the diet and mental health, the literature has not paid attention to possible relationships among sugar intake, personality traits, and cognitive–behavioural or emotional functioning. It is essential to learn about these mechanisms.

In health psychology, many studies confirm that an individual's personal resources and personality traits are essential for engaging in healthy behaviour [14]. And critical roles are played by stress-type personalities, the surrounding emotions, a sense of coherence, and self-efficacy [15,16]. Other studies emphasise the importance of an internal locus of control, emotional maturity, resilience to stress, autonomy, low levels of anxiety and fear, and high self-esteem [17,18]. Some researchers recognise that conscientiousness and agreeableness are associated with health-promoting behaviours, while neuroticism is associated with behaviours that harm health [19]. The critical mechanism underlying such relationships is the generation of positive affective states by conscientiousness and agreeableness and the generation of negative affective states by neuroticism [19]. The positive emotions that constitute the principal mechanism conducive for engaging in health-promoting eating behaviours promote health. Findings from other studies indicate the link among sugar consumption, the human brain, and human behaviour [7]. A link has been discovered between eating a diet rich in sugar and the occurrence of emotional disorders, such as anxiety and depression [11]. Some sources report that personality traits, such as neuroticism, extroversion, and conscientiousness, can influence preferences for sweet tastes [12]. Studies have shown that personality traits can, indeed, affect dietary choices, including the type of diet [13]. Individuals marked by neuroticism and alexithymia were likelier to have a low consumption of fruits and vegetables and an increased consumption of sugar and saturated fats [13]. Subjects characterised by neuroticism and extroversion consumed sweeter and saltier foods than people with conscientiousness characteristics [12]. An interesting observation was discovered for people who used stimulants, such as alcohol or drugs, in excess—in this group, the preference for sweet foods was high [20,21]. The association was stronger for those with a genetic predisposition to alcoholism [20,21]. Unfortunately, none of these studies comprehensively described the dietary patterns of those who prefer sweet tastes in their diet and their translation to psychological characteristics.

Given the lack of available information, a study was conducted to analyse sugar-related dietary patterns, personality traits, and cognitive–behavioural and emotional functioning as variables in eating behaviours. We hypothesise that there are correlations between selected character traits and sugar-related dietary patterns.

## 2. Study Sample and Methods

### 2.1. Study Design

This cross-sectional study was conducted among 624 working-age women. Informed consent was obtained from all the subjects involved in the study. The inclusion criterion was females aged 18–64. Exclusion criteria were (1) the age of the respondents (under 18 or over 64) and (2) the gender of the respondents. The subjects' flowchart through the study is shown in Figure 1. The characteristics of the collected study sample are presented in Section 3.1.

Data were collected between 2020 and 2021 using an anonymous Google Forms questionnaire. Recruitment was conducted using the snowball sampling method described previously, where new subjects were recruited by others to form a part of the sample [22]. All the procedures followed the ethical standards of the institutional and national research committees and the Helsinki Declaration. The participants consented to participate in the study with a digital informed consent form. The study was accepted by the local Institutional Review Board (Bioethical Commission at Poznan University of Medical Sciences, resolution no. 120/21).

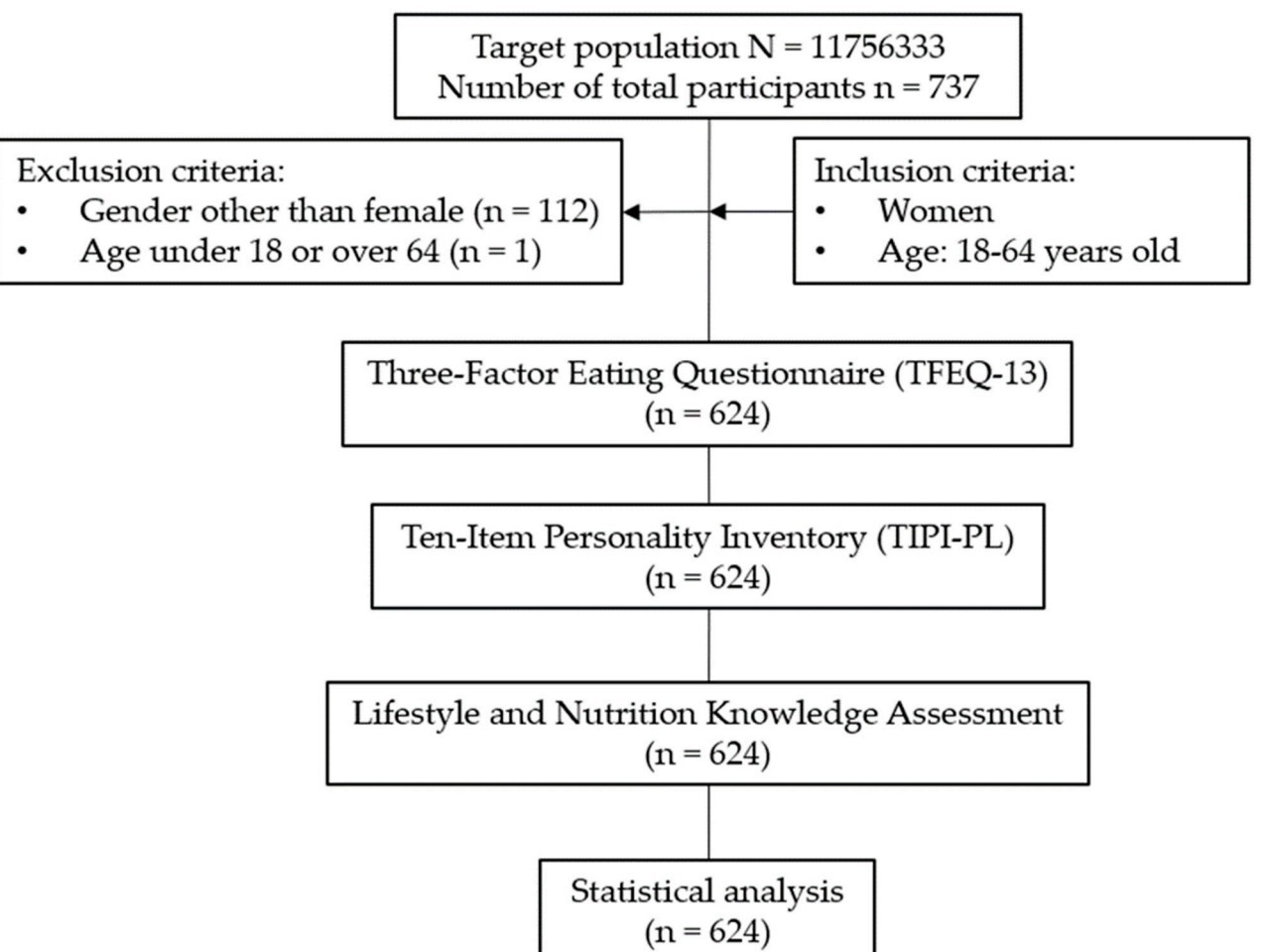

**Figure 1.** The subjects' flowchart through the study.

*2.2. Dietary Patterns (DPs)*

The frequency of the consumption was recorded using the lifestyle and nutrition knowledge assessment (KomPAN) [23,24]. The KomPAN questionnaire consists of four sections, within which it provides the following information: (1) dietary habits, (2) frequency of food consumption, (3) views on food and nutrition, and (4) lifestyle and personal information [24]. It was validated internationally, including for the Polish population between 15 and 65 years old [24–26]. The frequency of the food intake is shown in Table S1. The responses were converted to daily consumption frequencies, following the KomPAN procedure [23]. Dietary indexes were calculated for the established products and product groups, and the results were interpreted using the tercile division recommended previously. Two diet quality scores, namely, the pro-healthy diet index (pHDI-10) and the non-healthy diet index (nHDI-11), were determined using the frequency of the food intake [23,27,28]. In addition, the "sugar diet index" (sDI-7), which represents the dietary sugar intake relating to seven food groups, has also been created. All the index components are shown in Table 1.

The dietary patterns (DPs) were derived a posteriori using principal component analysis (PCA) with a varimax rotation. The input variables were the frequencies of the consumption of fruits, vegetables, fermented-milk drinks, cottage cheese, hard cheese, cured meat, buckwheat, oats, wholegrain pasta, legume-based foods, white bread and bakery products, butter, fried foods, sweetened beverages, and sweets. Other variables taken into account were sDI, nHDI, and pHDI. The data on the frequency of the food consumption were standardised. The sample size was sufficient to derive the DPs, as the ratio of the respondents to the input variables was 39:1 (624/16) [29,30].

**Table 1.** Characteristics of dietary indexes (pHDI-10, nHDI-11, and sDI-7).

| Food Group | Products Included |
| --- | --- |
| pHDI-10 pro-healthy diet index | (1) wholemeal bread; (2) buckwheat, oats, and whole-wheat pasta; (3) milk; (4) fermented-milk drinks; (5) cottage cheese; (6) white meat; (7) fish; (8) legume-based foods; (9) fruits; (10) vegetables |
| nHDI-11 non-healthy diet index | (1) white bread and bakery products; (2) white rice and pasta; (3) fast food; (4) fried food; (5) butter; (6) hard cheese; (7) red meat; (8) candies; (9) sweetened carbonated and non-carbonated drinks; (10) energy drinks; (11) alcoholic beverages |
| sDI-7 sugar diet index | (1) fruits; (2) fruit juices; (3) candies; (4) sweetened hot drinks; (5) sweetened carbonated or non-carbonated beverages; (6) energy drinks; (7) alcoholic beverages |

Three PCA-derived dietary patterns were identified. The sweet-Western dietary pattern (SWDP) was heavily loaded by the frequent consumption of white bread and bakery products, sweetened beverages, candies, cured meat, butter, and fried foods and high sDI and nHDI values. A pro-healthy dietary pattern (PHDP) reflected mainly the consumption of vegetables, fruits, legume-based foods, buckwheat, oats, and wholegrain pasta and high pHDI values. The consumption of cottage cheese, fermented-milk drinks, and hard cheese contributed heavily to the third pattern, the dairy dietary pattern (DDP). All the patterns explained 57% of the total variance; the shares in the variance explanation equalled 25%, 22%, and 10%, respectively, for the first, second, and third patterns. For further analyses, tercile intervals were calculated for each PCA-derived DP.

### 2.3. Personality Traits and Cognitive–Behavioural and Emotional Functioning

The three-factor eating questionnaire (TFEQ-13) and the ten-item personality inventory (TIPI-PL) were used to analyse the personality traits and cognitive–behavioural and emotional functioning of the study sample, respectively [31,32]. Both were validated internationally, including for the Polish population [33–43]. The TFEQ-13 distinguishes subjects whose behaviour towards diet was characterised by the cognitive restraint of eating (CR) subscale, which measures behaviours related to restricting the amount or type of food to control weight and body image (questions O1–O5). Uncontrolled eating (UE) measures the tendency to eat more than usual because of the loss of the control over eating or uncontrollable feelings of hunger that trigger an overeating attack (questions R1–R5). Emotional eating (EE) measures overeating episodes caused by feelings of lowered mood and anxiety (questions E1–E3) [32]. These three factors reproduced 56.8% of the variability in the entire set of observed variables. Cronbach's coefficient of internal consistency (Cronbach's alpha) for the whole scale was 0.78; for the subscales, it was 0.78, 0.76, and 0.72. Values are calculated separately for each subscale [32,44].

The TIPI inventory created by Gosling et al. was translated into a Polish adaptation by Sorokowska et al. [31,45]. The TIPI-PL consists of 10 statements that examine personality in five dimensions (neuroticism, extraversion, conscientiousness, openness to experience, and agreeableness) according to the five-factor model of personalities [46]. The person under examination is asked to respond using the phrase "I perceive myself as a person" to each statement, rating themselves on a Likert scale from 1 (strongly disagree) to 7 (strongly agree). The score is calculated separately for each dimension, calculating the average points awarded to the two relevant statements. The higher the average, the higher the intensity in the personality trait in question is [46].

The scores obtained by the respondents in the form of points were divided into terciles. The first tercile indicated a low intensity in the trait or eating behaviour. In contrast, the third tercile stood for a high intensity in the trait or eating behaviour.

### 2.4. Statistical Analysis

After considering the confidence level (98%) and the margin of error (3.63%), the calculated minimum sample size was 624 subjects [47]. Statistica v. 13.3 statistical software was used to calculate the sample size and study power [48]. The independent samples t-test and power analysis were carried out. The variables used in the calculations were pHDI and nHDI, the variables used to extract the feeding patterns. The minimum sample size for an adequate study power was 286; alpha = 0.05 and power = 95%. All the variables were checked for normality using the Kolmogorov–Smirnov test. The $\chi^2$ test was used to assess the distribution of the categorical variables. Principal component analysis (PCA) was used to isolate dietary patterns (DPs). The dietary patterns were identified by considering the following criteria: (1) the eigenvalues of the variable correlations >1.0, (2) the plot of the eigenvalues, and (3) the total explained variance [49]. Rotated factor loadings with an absolute value of |0.50| were considered to be specific to the pattern [49]. For each patient and each pattern, the scores were calculated as a product of the factor loading and food frequency consumption [49]. Next, for each dietary pattern, tertile intervals were calculated to measure the adherence to the patterns of each patient [49]. The logistic regression analysis searched for significant correlations among the obtained dietary patterns, trait intensities (TIPI-PLs), and factors (TFEQ-13s). The dependent variables were the three dietary patterns extracted from the studies performed, while the independent variables were the selected character traits that were analysed. Statistical analysis was also performed using Statistica v. 13.3 statistical software (StatSoft Polska Sp. z o.o. 2023) and Kit Plus version 5.0.96; available online: https://www.statsoft.pl/ (accessed on 14 December 2023) [48].

### 3. Results

### 3.1. Sociodemographic and Anthropometric Results of the Study Sample

Sociodemographic characteristics and anthropometric measurement results are shown in Table 2. Most of the sample was aged 18–26 (93%). Over half the women (73%) had a BMI indicating a normal body weight. More than 50% of the respondents admitted to living with their families. The most common place of residence (48%) was a city (>100,000 residents). The vast majority (62%) of the sample had an upper-secondary level of education. Also, the study profile was represented by humanities (34% humanities and psychology), technical (7%), and medical and nutrition (19% and 19%, respectively) courses.

**Table 2.** Characteristics of the study sample.

| Variable | Total Number of Participants N = 624 | | | |
|---|---|---|---|---|
| | **Mean** | **SD** | **Min.** | **Max.** |
| Weight (kg) | 61.0 | 10.3 | 41.0 | 104.0 |
| BMI (kg/m$^2$) | 21.8 | 3.4 | 15.6 | 37.5 |
| Age (years) | 22.7 | 4.5 | 18 | 54 |
| BMI interpretation: | n (%) | | | |
| → Underweight (<18.5 kg/m$^2$) | 73 (12) | | | |
| → Normal weight (≥18.5 kg/m$^2$ and <25 kg/m$^2$) | 459 (73) | | | |
| → Overweight (≥25 kg/m$^2$ and <30 kg/m$^2$) | 72 (12) | | | |
| → Obesity class I (≥30 kg/m$^2$ and <35 kg/m$^2$) | 17 (3) | | | |
| → Obesity class II (≥35 kg/m$^2$ and <40 kg/m$^2$) | 3 (0) | | | |
| → Obesity class III (≥40 kg/m$^2$) | 0 (0) | | | |
| Education level: | n (%) | | | |
| Upper secondary | 388 (62) | | | |
| BSc | 186 (30) | | | |
| MSc | 50 (8) | | | |

**Table 2.** *Cont.*

| Variable | Total Number of Participants N = 624 | | | |
|---|---|---|---|---|
| | **Mean** | **SD** | **Min.** | **Max.** |
| Major of study: | | n (%) | | |
|     Medical (e.g., medicine, midwifery, physiotherapy, and related fields) | | 117 (19) | | |
|     Nutrition | | 119 (19) | | |
|     Food technology | | 13 (2) | | |
|     Humanities and related fields | | 80 (13) | | |
|     Psychology/pedagogy and related fields | | 132 (21) | | |
|     Technical (e.g., polytechnics and related fields) | | 42 (7) | | |
|     Economics and related fields | | 121 (19) | | |
| Place of residence: | | | | |
|     City >100,000 inhabitants | | 300 (48) | | |
|     City 20–100,000 inhabitants | | 84 (13) | | |
|     City <20,000 inhabitants | | 74 (12) | | |
|     Village | | 166 (27) | | |
| Age: | | | | |
|     18–26 (years) | | 577 (93) | | |
|     27–35 (years) | | 27 (4) | | |
|     36–44 (years) | | 12 (2) | | |
|     45–54 (years) | | 8 (1) | | |
| Housing: | | | | |
|     I live with family. | | 342 (55) | | |
|     I live with a partner. | | 139 (22) | | |
|     I live with a roommate. | | 92 (15) | | |
|     I live alone. | | 51 (8) | | |

### 3.2. Dietary Patterns

Three DPs were identified: "sweet-Western DP" (SWDP), "pro-healthy DP" (PHDP), and "dairy DP" (DDP). Table 3 shows the factor loadings of the 16 indicators used to extract the 3 dietary patterns. A high-frequency intake of sugar sources, sweetened beverages, candies, butter, fried foods, cured meat, and white bread and bakery products characterised the SWDP. This SWDP was also marked by high nHDI values. The PHDP was related to a high-frequency intake of vegetables, legume-based foods, fruits, buckwheat, oats, and wholegrain pasta. This PHDP was also characterised by high pHDI values. The DDP was characterised by a high-frequency intake of fermented-milk drinks, cottage cheese, and hard cheese. The food frequency intake of the study group described in means and medians is available in Table S2. In contrast, the mean food frequency intakes per day for those with high, moderate, and low adherences to the sweet-Western dietary pattern (SWDP), pro-healthy dietary pattern (PHDP), and dairy dietary pattern (DDP) are shown in Table S3, Table S4, and Table S5, respectively.

The adherences to the sweet-Western dietary pattern (SWDP) and pro-healthy dietary pattern (PHDP) and their relations to the examined traits are presented in Table 4. The severity of the trait conscientiousness reduced the likelihood of high intakes of sources of simple sugars in the diet by 34%. The intensity in behaviours such as cognitive restraint (CR) was found to increase, by almost 2 times, the probability of a high adherence to the SWDP, associated with an intensification in the intake of sources of simple sugars. Moreover, it was found that intensifying a behaviour, such as emotional eating (EE), by 37% reduces the chance of a high adherence to non-healthy dietary behaviours.

**Table 3.** Factor loadings of three dietary patterns (DPs): sweet-Western (SWDP), pro-healthy (PHDP), and dairy (DDP).

|  | SWDP | PHDP | DDP |
|---|---|---|---|
| nHDI | 0.94 | −0.07 | 0.22 |
| Vegetables | −0.01 | 0.87 | 0.02 |
| Fruits | 0.12 | 0.88 | 0.01 |
| pHDI | 0.06 | 0.83 | 0.49 |
| Cottage cheese | 0.06 | 0.16 | 0.82 |
| Fermented-milk drinks | 0.01 | 0.32 | 0.73 |
| sDI | 0.73 | 0.39 | −0.02 |
| White bread and bakery products | 0.63 | −0.07 | −0.03 |
| Candies | 0.61 | 0.17 | −0.18 |
| Cured meat | 0.61 | −0.14 | 0.26 |
| Butter | 0.59 | −0.12 | 0.18 |
| Legume-based foods | −0.14 | 0.58 | 0.02 |
| Hard cheese | 0.39 | −0.13 | 0.58 |
| Sweetened beverages | 0.58 | 0.00 | 0.07 |
| Fried foods | 0.54 | −0.13 | 0.00 |
| Buckwheat, oats, and wholegrain pasta | −0.21 | 0.53 | 0.34 |

Factor loadings greater than 0.50 are highlighted with a gray background.

**Table 4.** The adherences to the sweet-Western dietary pattern (SWDP) and pro-healthy dietary pattern (PHDP) and their relations to the examined features.

|  | High Adherence to SWDP | | Moderate Adherence to SWDP | | Low Adherence to SWDP | |
|---|---|---|---|---|---|---|
|  | n | OR (CI 95%), $p$ | n | OR (CI 95%), $p$ | n | OR (CI 95%), $p$ |
| Extraversion 3rd tercile [1] | 78 | 1.03 (0.73; 1.44), $p = 0.88$ [3] $p = 0.91$ [4] | 63 | 0.71 (0.50; 1.02), $p = 0.06$ [3] $p = 0.17$ [4] | 85 | 1.35 (0.96; 1.90), $p = 0.09$ [3] $p = 0.19$ [4] |
| Extraversion 3rd tercile | 78 | 1.02 (0.72; 1.45), $p = 0.89$ [3] $p = 0.89$ [4] | 63 | 0.71 (0.50; 1.02), $p = 0.06$ [3] $p = 0.16$ [4] | 85 | 1.36 (0.96; 1.91), $p = 0.08$ [3] $p = 0.20$ [4] |
| Agreeableness 3rd tercile [1] | 108 | 0.87 (0.62; 1.21), $p = 0.40$ [3] $p = 0.56$ [4] | 107 | 0.98 (0.70; 1.36), $p = 0.91$ [3] $p = 0.91$ [4] | 116 | 1.18 (0.84; 1.65), $p = 0.33$ [3] $p = 0.54$ [4] |
| Agreeableness 3rd tercile | 108 | 0.86 (0.62; 1.20), $p = 0.38$ [3] $p = 0.51$ [4] | 107 | 0.97 (0.70; 1.36), $p = 0.87$ [3] $p = 0.89$ [4] | 116 | 1.19 (0.85; 1.67), $p = 0.30$ [3] $p = 0.49$ [4] |
| Conscientiousness 3rd tercile [1] | 91 | 0.67 (0.48; 0.94), $p < 0.05$ *,[3] $p = 0.08$ [4] | 103 | 1.10 (0.78; 1.53), $p = 0.60$ [3] $p = 0.75$ [4] | 113 | 1.36 (0.97; 1.90), $p = 0.07$ [3] $p = 0.17$ [4] |
| Conscientiousness 3rd tercile | 91 | 0.66 (0.47; 0.93), $p < 0.05$ *,[3] $p = 0.08$ [4] | 103 | 1.08 (0.77; 1.52), $p = 0.65$ [3] $p = 0.81$ [4] | 113 | 1.40 (1.00; 1.97), $p = 0.05$ [3] $p = 0.15$ [4] |
| Emotional Stability 3rd tercile [1] | 91 | 1.07 (0.76; 1.49), $p = 0.70$ [3] $p = 0.83$ [4] | 78 | 0.82 (0.58; 1.16), $p = 0.25$ [3] $p = 0.47$ [4] | 91 | 1.14 (0.81; 1.59), $p = 0.46$ [3] $p = 0.61$ [4] |
| Emotional Stability 3rd tercile [2] | 91 | 1.07 (0.76; 1.50), $p = 0.71$ [3] $p = 0.81$ [4] | 78 | 0.80 (0.56; 1.13), $p = 0.20$ [3] $p = 0.40$ [4] | 91 | 1.17 (0.83; 1.64), $p = 0.37$ [3] $p = 0.51$ [4] |
| Openness to Experiences 3rd tercile [1] | 94 | 0.86 (0.62; 1.20), $p = 0.38$ [3] $p = 0.54$ [4] | 93 | 0.95 (0.68; 1.33), $p = 0.78$ [3] $p = 0.85$ [4] | 104 | 1.23 (0.88; 1.71), $p = 0.23$ [3] $p = 0.47$ [4] |
| Openness to Experiences 3rd tercile [2] | 94 | 0.86 (0.62; 1.20), $p = 0.37$ [3] $p = 0.51$ [4] | 93 | 0.96 (0.69; 1.35), $p = 0.82$ [3] $p = 0.89$ [4] | 104 | 1.21 (0.87; 1.70), $p = 0.26$ [3] $p = 0.47$ [4] |

**Table 4.** *Cont.*

| | High Adherence to SWDP | | Moderate Adherence to SWDP | | Low Adherence to SWDP | |
|---|---|---|---|---|---|---|
| | **n** | **OR (CI 95%),** $p$ | **n** | **OR (CI 95%),** $p$ | **n** | **OR (CI 95%),** $p$ |
| Uncontrolled Eating (UE) 3rd tercile [1] | 101 | 0.85 (0.61; 1.18), $p = 0.32$ [3] $p = 0.54$ [4] | 90 | 0.71 (0.51; 0.99), $p < 0.05$ *,[3] $p = 0.13$ [4] | 122 | 1.67 (1.19; 2.34), $p < 0.01$ *,[3] $p < 0.05$ *,[4] |
| Uncontrolled Eating (UE) 3rd tercile [2] | 101 | 0.84 (0.60; 1.17), $p = 0.30$ [3] $p = 0.49$ [4] | 90 | 0.70 (0.50; 0.99), $p < 0.05$ *,[3] $p = 0.14$ [4] | 122 | 1.70 (1.21; 2.39), $p < 0.01$ *,[3] $p < 0.05$ *,[4] |
| Cognitive Restraint (CR) 3rd tercile [1] | 118 | 1.93 (1.38; 2.70), $p < 0.001$ *,[3] $p < 0.01$ *,[4] | 106 | 1.57 (1.12; 2.20), $p < 0.01$ *,[3] $p < 0.05$ *,[4] | 55 | 0.31 (0.21; 0.44), $p < 0.001$ *,[3] $p < 0.001$ *,[4] |
| Cognitive Restraint (CR) 3rd tercile [2] | 118 | 1.93 (1.38; 2.70), $p < 0.001$ *,[3] $p < 0.01$ *,[4] | 106 | 1.59 (1.13; 2.23), $p < 0.01$ *,[3] $p < 0.05$ *,[4] | 55 | 0.30 (0.21; 0.44), $p < 0.001$ *,[3] $p < 0.001$ *,[4] |
| Emotional Eating (EE) 3rd tercile [1] | 101 | 0.64 (0.46; 0.90), $p < 0.01$ *,[3] $p < 0.05$ *,[4] | 113 | 1.06 (0.76; 1.49), $p = 0.72$ [3] $p = 0.83$ [4] | 127 | 1.48 (1.05; 2.08), $p < 0.05$ *,[3] $p = 0.08$ [4] |
| Emotional Eating (EE) 3rd tercile [2] | 101 | 0.63 (0.45; 0.88), $p < 0.01$ *,[3] $p < 0.05$ *,[4] | 113 | 1.07 (0.76; 1.51), $p = 0.70$ [3] $p = 0.81$ [4] | 127 | 1.51 (1.07; 2.13), $p < 0.05$ *,[3] $p = 0.08$ [4] |
| | High Adherence to PHDP | | Moderate Adherence to PHDP | | Low Adherence to PHDP | |
| Extraversion 3rd tercile [1] | 85 | 1.27 (0.90; 1.79), $p = 0.17$ [3] $p = 0.29$ [4] | 65 | 0.74 (0.52; 1.05), $p = 0.09$ [3] $p = 0.21$ [4] | 76 | 1.06 (0.75; 1.49), $p = 0.76$ [3] $p = 0.86$ [4] |
| Extraversion 3rd tercile [2] | 85 | 1.24 (0.88; 1.76), $p = 0.21$ [3] $p = 0.34$ [4] | 65 | 0.74 (0.52; 1.05), $p = 0.09$ [3] $p = 0.23$ [4] | 76 | 1.09 (0.77; 1.54), $p = 0.64$ [3] $p = 0.74$ [4] |
| Agreeableness 3rd tercile [1] | 122 | 1.30 (0.93; 1.81), $p = 0.13$ [3] $p = 0.26$ [4] | 106 | 0.91 (0.65; 1.27), $p = 0.58$ [3] $p = 0.69$ [4] | 103 | 0.85 (0.61; 1.18), $p = 0.33$ [3] $p = 0.44$ [4] |
| Agreeableness 3rd tercile [2] | 122 | 1.25 (0.89; 1.75), $p = 0.20$ [3] $p = 0.34$ [4] | 106 | 0.91 (0.65; 1.28), $p = 0.60$ [3] $p = 0.72$ [4] | 103 | 0.88 (0.62; 1.23), $p = 0.44$ [3] $p = 0.55$ [4] |
| Conscientiousness 3rd tercile [1] | 127 | 1.90 (1.35; 2.65), $p < 0.001$ *,[3] $p < 0.01$ *,[4] | 90 | 0.72 (0.51; 1.01), $p = 0.05$ [3] $p = 0.16$ [4] | 90 | 0.73 (0.52; 1.02), $p = 0.07$ [3] $p = 0.17$ [4] |
| Conscientiousness 3rd tercile [2] | 127 | 1.80 (1.28; 2.53), $p < 0.001$ *,[3] $p < 0.01$ *,[4] | 90 | 0.72 (0.52; 1.01), $p = 0.06$ [3] $p = 0.18$ [4] | 90 | 0.77 (0.55; 1.08), $p = 0.13$ [3] $p = 0.26$ [4] |
| Emotional Stability 3rd tercile [1] | 83 | 0.84 (0.60; 1.18), $p = 0.33$ [3] $p = 0.44$ [4] | 87 | 1.04 (0.74; 1.46), $p = 0.84$ [3] $p = 0.87$ [4] | 90 | 1.15 (0.82; 1.61), $p = 0.43$ [3] $p = 0.54$ [4] |
| Emotional Stability 3rd tercile [2] | 83 | 0.82 (0.58; 1.15), $p = 0.25$ [3] $p = 0.38$ [4] | 87 | 1.05 (0.75; 1.48), $p = 0.78$ [3] $p = 0.83$ [4] | 90 | 1.17 (0.83; 1.65), $p = 0.38$ [3] $p = 0.50$ [4] |
| Openness to Experiences 3rd tercile [1] | 108 | 1.28 (0.92; 1.79), $p = 0.14$ [3] $p = 0.26$ [4] | 97 | 1.03 (0.74; 1.43), $p = 0.87$ [3] $p = 0.87$ [4] | 86 | 0.75 (0.54; 1.06), $p = 0.10$ [3] $p = 0.22$ [4] |
| Openness to Experiences 3rd tercile [2] | 108 | 1.30 (0.93; 1.81), $p = 0.13$ [3] $p = 0.26$ [4] | 97 | 1.02 (0.73; 1.43), $p = 0.90$ [3] $p = 0.90$ [4] | 86 | 0.75 (0.53; 1.05), $p = 0.10$ [3] $p = 0.23$ [4] |
| Uncontrolled Eating (UE) 3rd tercile [1] | 126 | 1.74 (1.24; 2.43), $p < 0.01$ *,[3] $p < 0.01$ *,[4] | 97 | 0.83 (0.60; 1.16), $p = 0.28$ [3] $p = 0.42$ [4] | 90 | 0.69 (0.49; 0.96), $p < 0.05$ *,[3] $p = 0.10$ [4] |

**Table 4.** *Cont.*

| | High Adherence to SWDP | | Moderate Adherence to SWDP | | Low Adherence to SWDP | |
|---|---|---|---|---|---|---|
| | n | OR (CI 95%), *p* | n | OR (CI 95%), *p* | n | OR (CI 95%), *p* |
| Uncontrolled Eating (UE) 3rd tercile [2] | 126 | 1.66 (1.19; 2.34), $p < 0.01$ *,[3] $p < 0.05$ *,[4] | 97 | 0.83 (0.60; 1.17), $p = 0.29$ [3] $p = 0.41$ [4] | 90 | 0.72 (0.51; 1.01), $p = 0.06$ [3] $p = 0.18$ [4] |
| Cognitive Restraint (CR) 3rd tercile [1] | 67 | 0.43 (0.30; 0.61), $p < 0.001$ *,[3] $p < 0.001$ *,[4] | 114 | 1.90 (1.36; 2.67), $p < 0.001$ *,[3] $p < 0.01$ *,[4] | 98 | 1.20 (0.86; 1.69), $p = 0.28$ [3] $p = 0.42$ [4] |
| Cognitive Restraint (CR) 3rd tercile [2] | 67 | 0.40 (0.28; 0.58), $p < 0.001$ *,[3] $p < 0.001$ *,[4] | 114 | 1.91 (1.36; 2.68), $p < 0.001$ *,[3] $p < 0.01$ *,[4] | 98 | 1.25 (0.89; 1.76), $p = 0.19$ [3] $p = 0.34$ [4] |
| Emotional Eating (EE) 3rd tercile [1] | 132 | 1.58 (1.12; 2.21), $p < 0.01$ *,[3] $p < 0.05$ *,[4] | 111 | 0.96 (0.68; 1.34), $p = 0.79$ [3] $p = 0.86$ [4] | 98 | 0.66 (0.47; 0.93), $p < 0.05$*,[3] $p = 0.07$ [4] |
| Emotional Eating (EE) 3rd tercile [2] | 132 | 1.47 (1.04; 2.08), $p < 0.05$ *,[3] $p = 0.13$ [4] | 111 | 0.96 (0.68; 1.34), $p = 0.79$ [3] $p = 0.83$ [4] | 98 | 0.71 (0.50; 1.00), $p < 0.05$ *,[3] $p = 0.18$ [4] |

The *p* values below the statistical significance threshold are marked with an asterisk (*) ($p < 0.05$). [1]—Logistic regression results before adjustment. [2]—Logistic regression results adjusted by BMI and age of the study participants. [3]—The value of "*p*" before the Benjamini–Hochberg correction. [4]—The value of "*p*" after the Benjamini–Hochberg correction.

The severity of the conscientiousness trait increased the probability of the high consumption of pro-healthy products by as much as 80%. This study showed that the severity of the uncontrolled eating (UE) trait increased the likelihood of the increased adherence to the PHDP by 66%. It was revealed that intensifying behaviours such as cognitive restraint (CR) reduced the probability of the high adherence to pro-healthy dietary behaviours by 60%. Moreover, it was found that intensifying a behaviour such as emotional eating (EE) by 47% increased the chance of a high adherence to the PHDP.

The adherence to the dairy dietary pattern (DDP) and its relation to the examined traits are presented in Table 5. The severity of the trait extraversion reduced the probability of a low adherence to the DDP by as much as 33%. In comparison, the severity of the introversion (low extraversion) reduced the likelihood of a high adherence to the DDP by 30%.

**Table 5.** The adherence to the dairy dietary pattern (DDP) and its relation to the examined features.

| | High Adherence to DDP | | Moderate Adherence to DDP | | Low Adherence to DDP | |
|---|---|---|---|---|---|---|
| | n | OR (CI 95%), *p* | n | OR (CI 95%), *p* | n | OR (CI 95%), *p* |
| Extraversion 3rd tercile [1] | 86 | 1.33 (0.94; 1.87), $p = 0.11$ [3] $p = 0.73$ [4] | 78 | 1.10 (0.78; 1.55), $p = 0.59$ [3] $p = 0.86$ [4] | 62 | 0.67 (0.47; 0.96), $p < 0.05$ *,[3] $p = 0.67$ [4] |
| Extraversion 3rd tercile [2] | 86 | 1.32 (0.93; 1.85), $p = 0.12$ [3] $p = 0.76$ [4] | 78 | 1.12 (0.79; 1.58), $p = 0.52$ [3] $p = 0.81$ [4] | 62 | 0.67 (0.47; 0.96), $p < 0.05$ *,[3] $p = 0.70$ [4] |
| Extraversion 1st tercile (Introversion) [1] | 65 | 0.70 (0.49; 0.99), $p < 0.05$ *,[3] $p = 0.67$ [4] | 80 | 1.18 (0.84; 1.70), $p = 0.34$ [3] $p = 0.73$ [4] | 80 | 1.21 (0.86; 1.71), $p = 0.28$ [3] $p = 0.73$ [4] |
| Extraversion 1st tercile (Introversion) [2] | 65 | 0.70 (0.50; 1.00), $p < 0.05$ *,[3] $p = 0.70$ [4] | 80 | 1.18 (0.83; 1.66), $p = 0.36$ [3] $p = 0.81$ [4] | 80 | 1.21 (0.86; 1.71), $p = 0.28$ [3] $p = 0.81$ [4] |
| Agreeableness 3rd tercile [1] | 109 | 0.91 (0.65; 1.26), $p = 0.56$ [3] $p = 0.86$ [4] | 111 | 1.04 (0.74; 1.45), $p = 0.84$ [3] $p = 0.91$ [4] | 111 | 1.07 (0.76; 1.50), $p = 0.70$ [3] $p = 0.88$ [4] |

**Table 5.** *Cont.*

| | High Adherence to DDP | | Moderate Adherence to DDP | | Low Adherence to DDP | |
|---|---|---|---|---|---|---|
| | n | OR (CI 95%), *p* | n | OR (CI 95%), *p* | n | OR (CI 95%), *p* |
| Agreeableness 3rd tercile [2] | 109 | 0.89 (0.64; 1.25), *p* = 0.50 [3] *p* = 0.81 [4] | 111 | 1.06 (0.76; 1.49), *p* = 0.72 [3] *p* = 0.94 [4] | 111 | 1.06 (0.75; 1.48), *p* = 0.75 [3] *p* = 0.94 [4] |
| Conscientiousness 3rd tercile [1] | 110 | 1.18 (0.84; 1.64), *p* = 0.34 [3] *p* = 0.73 [4] | 99 | 0.92 (0.66; 1.29), *p* = 0.63 [3] *p* = 0.86 [4] | 98 | 0.92 (0.66; 1.29), *p* = 0.63 [3] *p* = 0.86 [4] |
| Conscientiousness 3rd tercile [2] | 110 | 1.15 (0.82; 1.62), *p* = 0.41 [3] *p* = 0.81 [4] | 99 | 0.96 (0.69; 1.35), *p* = 0.82 [3] *p* = 0.95 [4] | 98 | 0.90 (0.64; 1.26), *p* = 0.54 [3] *p* = 0.81 [4] |
| Emotional Stability 3rd tercile [1] | 98 | 1.33 (0.95; 1.86), *p* = 0.10 [3] *p* = 0.73 [4] | 80 | 0.83 (0.59; 1.17), *p* = 0.28 [3] *p* = 0.73 [4] | 82 | 0.90 (0.64; 1.27), *p* = 0.56 [3] *p* = 0.86 [4] |
| Emotional Stability 3rd tercile [2] | 98 | 1.33 (0.94; 1.88), *p* = 0.10 [3] *p* = 0.76 [4] | 80 | 0.84 (0.59; 1.18), *p* = 0.32 [3] *p* = 0.81 [4] | 82 | 0.89 (0.63; 1.26), *p* = 0.51 [3] *p* = 0.81 [4] |
| Emotional Stability 1st tercile (Neuroticism) [1] | 80 | 0.78 (0.56; 1.10), *p* = 0.15 [3] *p* = 0.73 [4] | 87 | 1.02 (0.72; 1.44), *p* = 0.90 [3] *p* = 0.91 [4] | 93 | 1.25 (0.89; 1.76), *p* = 0.19 [3] *p* = 0.73 [4] |
| Emotional Stability 1st tercile (Neuroticism) [2] | 80 | 0.78 (0.55; 1.09), *p* = 0.15 [3] *p* = 0.76 [4] | 87 | 1.02 (0.73; 1.43), *p* = 0.92 [3] *p* = 0.95 [4] | 93 | 1.26 (0.90; 1.78), *p* = 0.18 [3] *p* = 0.76 [4] |
| Openness to Experiences 3rd tercile [1] | 102 | 1.09 (0.78; 1.53), *p* = 0.60 [3] *p* = 0.86 [4] | 101 | 1.14 (0.82; 1.59), *p* = 0.45 [3] *p* = 0.84 [4] | 88 | 0.80 (0.57; 1.12), *p* = 0.20 [3] *p* = 0.73 [4] |
| Openness to Experiences 3rd tercile [2] | 102 | 1.10 (0.79; 1.53), *p* = 0.59 [3] *p* = 0.85 [4] | 101 | 1.14 (0.81;1.59), *p* = 0.46 [3] *p* = 0.81 [4] | 88 | 0.80 (0.57; 1.13), *p* = 0.20 [3] *p* = 0.76 [4] |
| Uncontrolled Eating (UE) 3rd tercile [1] | 107 | 1.02 (0.73; 1.42), *p* = 0.91 [3] *p* = 0.91 [4] | 102 | 0.95 (0.68; 1.33), *p* = 0.76 [3] *p* = 0.91 [4] | 104 | 1.04 (0.74; 1.45), *p* = 0.84 [3] *p* = 0.91 [4] |
| Uncontrolled Eating (UE) 3rd tercile [2] | 107 | NS | 102 | 0.98 (0.70; 1.38), *p* = 0.91 [3] *p* = 0.95 [4] | 104 | 1.02 (0.73; 1.43), *p* = 0.90 [3] *p* = 0.95 [4] |
| Cognitive Restraint (CR) 3rd tercile [1] | 89 | 0.85 (0.61; 1.18), *p* = 0.33 [3] *p* = 0.73 [4] | 90 | 0.93 (0.66; 1.30), *p* = 0.66 [3] *p* = 0.86 [4] | 100 | 1.28 (0.91; 1.79), *p* = 0.15 [3] *p* = 0.73 [4] |
| Cognitive Restraint (CR) 3rd tercile [2] | 89 | 0.83 (0.59; 1.16), *p* = 0.28 [3] *p* = 0.81 [4] | 90 | 0.95 (0.68; 1.33), *p* = 0.75 [3] *p* = 0.94 [4] | 100 | 1.27 (0.91; 1.78), *p* = 0.16 [3] *p* = 0.76 [4] |
| Emotional Eating (EE) 3rd tercile [1] | 122 | 1.20 (0.86; 1.67) *p* = 0.30 [3] *p* = 0.73 [4] | 112 | 0.97 (0.69; 1.36), *p* = 0.85 [3] *p* = 0.91 [4] | 107 | 0.86 (0.62; 1.21), *p* = 0.39 [3] *p* = 0.78 [4] |
| Emotional Eating (EE) 3rd tercile [2] | 122 | 1.16 (0.83; 1.63), *p* = 0.39 [3] *p* = 0.81 [4] | 112 | 1.02 (0.73; 1.43), *p* = 0.91 [3] *p* = 0.95 [4] | 107 | 0.84 (0.60; 1.19), *p* = 0.33 [3] *p* = 0.81 [4] |

The *p* values below the statistical significance threshold are marked with an asterisk (*) ($p < 0.05$). [1]—Logistic regression results before adjustment. [2]—Logistic regression results adjusted by BMI and age of the study participants. [3]—The value of "*p*" before the Benjamini–Hochberg correction. [4]—The value of "*p*" after the Benjamini–Hochberg correction.

### 3.3. Ten-Item Personality Inventory (TIPI-PL) Results

The results of the TIPI-PL questionnaire are shown in Table S6. The vast majority of the female respondents confirm that they see themselves as being extroverted (68%), dependable (72%), open to new experiences (77%), sympathetic (86%), and organised (72%).

More than half the group experiences anxiety (59%), while nearly 60% of the women surveyed do not consider themselves as being emotionally stable. Moreover, more than half (51%) the group does not think they are characterised by quietness and reservedness, and nearly 40% of the female respondents believe they are uncreative. Most of the group (65%) does not consider themselves as being quarrelsome.

The intensities of the features examined through the TIPI-PL questionnaire are shown in Table S7. A tercile division was performed on these values. The results present that the majority of the group had low levels of agreeableness (40%), conscientiousness (42%), and emotional stability (42%).

Almost half the group (44%) was characterised by a medium intensity in extraversion. In the case of openness to experiences, there was a fairly even split, with a weak advantage for low intensity—37% of the group.

### 3.4. Three-Factor Eating Questionnaire (TFEQ-13) Results

The interpreted outcomes of the questionnaire are shown in Table S8. Almost half (45%) the study sample had a low intensity in emotional eating. In the cases of uncontrolled eating and cognitive restraint, a medium or high level of the disorder was observed in more than 60% of the respondents.

The adherences to uncontrolled eating (UE), cognitive restraint (CR), and emotional eating (EE) and their relations to the examined traits are presented in Table 6. Increased extraversion and agreeableness decreased the probability of a low adherence to the UE by 35% and 30%, respectively. A high intensity in conscientiousness increased the chance of a high adherence to uncontrolled eating by as much as 74%. Enhanced emotional stability increased the chance of a moderate adherence to the UE by 51%. In comparison, the severity of the neuroticism increased the likelihood of a low adherence to uncontrolled eating (UE) by as much as 75%. The conscientiousness trait reduced the probability of a high adherence to cognitive restraint (CR) by 36%.

**Table 6.** The adherences to uncontrolled eating (UE), cognitive restraint (CR), and emotional eating (EE) and their relations to the examined features.

| | High Adherence to UE | | Moderate Adherence to UE | | Low Adherence to UE | |
|---|---|---|---|---|---|---|
| | **n** | **OR (CI 95%), *p*** | **n** | **OR (CI 95%), *p*** | **n** | **OR (CI 95%), *p*** |
| Extraversion 3rd tercile [1] | 83 | 1.24 (0.88; 1.75); $p = 0.22$ [3] $p = 0.31$ [4] | 74 | 1.32 (0.93; 1.89), $p = 0.12$ [3] $p = 0.22$ [4] | 69 | 0.63 (0.44; 0.89), $p < 0.01$ *,[3] $p < 0.05$ *,[4] |
| Extraversion 3rd tercile [2] | 83 | 1.21 (0.85; 1.71), $p = 0.29$ [3] $p = 0.38$ [4] | 74 | 1.32 (0.92; 1.89), $p = 0.13$ [3] $p = 0.23$ [4] | 69 | 0.65 (0.46; 0.92), $p < 0.05$ *,[3] $p = 0.05$ [4] |
| Agreeableness 3rd tercile [1] | 120 | 1.28 (0.92; 1.79), $p = 0.14$ [3] $p = 0.24$ [4] | 102 | 1.21 (0.85; 1.71), $p = 0.29$ [3] $p = 0.37$ [4] | 109 | 0.67 (0.48; 0.93), $p < 0.05$ *,[3] $p < 0.05$ *,[4] |
| Agreeableness 3rd tercile [2] | 120 | 1.22 (0.87; 1.71), $p = 0.26$ [3] $p = 0.38$ [4] | 102 | 1.20 (0.85; 1.71), $p = 0.30$ [3] $p = 0.38$ [4] | 109 | 0.70 (0.50; 0.98), $p < 0.05$ *,[3] $p = 0.08$ [4] |
| Conscientiousness 3rd tercile [1] | 125 | 1.88 (1.34; 2.63), $p < 0.001$ *,[3] $p < 0.01$ *,[4] | 94 | 1.17 (0.83; 1.65), $p = 0.38$ [3] $p = 0.46$ [4] | 88 | 0.48 (0.34; 0.67), $p < 0.001$ *,[3] $p < 0.001$ *,[4] |
| Conscientiousness 3rd tercile [2] | 125 | 1.74 (1.23; 2.45), $p < 0.01$ *,[3] $p < 0.01$ *,[4] | 94 | 1.16 (0.82; 1.65), $p = 0.40$ [3] $p = 0.48$ [4] | 88 | 0.51 (0.37; 0.72), $p < 0.001$ *,[3] $p < 0.01$ *,[4] |
| Emotional Stability 3rd tercile [1] | 99 | 1.40 (1.00; 1.96), $p < 0.05$ *,[3] $p = 0.10$ [4] | 88 | 1.49 (1.05; 2.11), $p < 0.05$ *,[3] $p = 0.06$ [4] | 73 | 0.50 (0.35; 0.70), $p < 0.001$ *,[3] $p < 0.001$ *,[4] |

**Table 6.** *Cont.*

| | High Adherence to UE | | Moderate Adherence to UE | | Low Adherence to UE | |
|---|---|---|---|---|---|---|
| | **n** | **OR (CI 95%),** $p$ | **n** | **OR (CI 95%),** $p$ | **n** | **OR (CI 95%),** $p$ |
| Emotional Stability 3rd tercile [2] | 99 | 1.36 (0.96; 1.91), $p = 0.08$ [3] $p = 0.16$ [4] | 88 | 1.51 (1.06; 2.14), $p < 0.05$ *,[3] $p = 0.06$ [4] | 73 | 0.51 (0.36; 0.72), $p < 0.001$ *,[3] $p < 0.01$ *,[4] |
| Emotional Stability 1st tercile (Neuroticism) [1] | 80 | 0.80 (0.57; 1.12), $p = 0.20$ [3] $p = 0.30$ [4] | 63 | 0.67 (0.47; 0.96), $p < 0.05$ *,[3] $p = 0.06$ [4] | 117 | 1.75 (1.26; 2.43), $p < 0.001$ *,[3] $p < 0.001$ *,[4] |
| Emotional Stability 1st tercile (Neuroticism) [2] | 80 | 0.81 (0.57; 1.15), $p = 0.23$ [3] $p = 0.38$ [4] | 63 | 0.66 (0.46; 0.95), $p < 0.05$ *,[3] $p = 0.06$ [4] | 117 | 1.75 (1.25; 2.45), $p < 0.01$ *,[3] $p < 0.01$ *,[4] |
| Openness to Experiences 3rd tercile [1] | 100 | 1.06 (0.76; 1.48), $p = 0.73$ [3] $p = 0.73$ [4] | 80 | 0.87 (0.62; 1.23), $p = 0.44$ [3] $p = 0.49$ [4] | 111 | 1.07 (0.77; 1.48), $p = 0.70$ [3] $p = 0.73$ [4] |
| Openness to Experiences 3rd tercile [2] | 100 | 1.08 (0.77; 1.51), $p = 0.66$ [3] $p = 0.69$ [4] | 80 | 0.87 (0.61; 1.23), $p = 0.43$ [3] $p = 0.48$ [4] | 111 | 1.05 (0.76; 1.47), $p = 0.76$ [3] $p = 0.76$ [4] |
| | High Adherence to CR | | Moderate Adherence to CR | | Low Adherence to CR | |
| Extraversion 3rd tercile [1] | 65 | 1.02 (0.70; 1.49), $p = 0.93$ [3] $p = 0.93$ [4] | 73 | 0.82 (0.58; 1.16), $p = 0.27$ [3] $p = 0.79$ [4] | 88 | 1.19 (0.85; 1.67), $p = 0.32$ [3] $p = 0.79$ [4] |
| Extraversion 3rd tercile [2] | 65 | 1.00 (0.74; 1.35), $p = 0.99$ [3] $p = 0.99$ [4] | 73 | 0.83 (0.58; 1.17), $p = 0.28$ [3] $p = 0.71$ [4] | 88 | 1.21 (0.86; 1.69), $p = 0.28$ [3] $p = 0.71$ [4] |
| Agreeableness 3rd tercile [1] | 92 | 0.93 (0.65; 1.31), $p = 0.67$ [3] $p = 0.89$ [4] | 118 | 1.05 (0.76; 1.46), $p = 0.76$ [3] $p = 0.89$ [4] | 121 | 1.02 (0.71; 1.45), $p = 0.92$ [3] $p = 0.93$ [4] |
| Agreeableness 3rd tercile [2] | 92 | 0.91 (0.64; 1.29), $p = 0.59$ [3] $p = 0.80$ [4] | 118 | 1.06 (0.76; 1.48), $p = 0.72$ [3] $p = 0.90$ [4] | 121 | 1.03 (0.74; 1.43), $p = 0.88$ [3] $p = 0.94$ [4] |
| Conscientiousness 3rd tercile [1] | 75 | 0.67 (0.47; 0.96), $p < 0.05$ *,[3] $p = 0.20$ [4] | 106 | 0.95 (0.69; 1.32), $p = 0.77$ [3] $p = 0.89$ [4] | 126 | 1.49 (1.07; 2.07), $p < 0.05$ *,[3] $p = 0.20$ [4] |
| Conscientiousness 3rd tercile [2] | 75 | 0.64 (0.45; 0.92), $p < 0.05$ *,[3] $p = 0.11$ [4] | 106 | 0.97 (0.69; 1.35), $p = 0.83$ [3] $p = 0.94$ [4] | 126 | 1.53 (1.09; 2.14), $p < 0.05$ *,[3] $p = 0.11$ [4] |
| Emotional Stability 3rd tercile [1] | 77 | 1.10 (0.77; 1.56), $p = 0.61$ [3] $p = 0.89$ [4] | 95 | 1.11 (0.80; 1.56), $p = 0.52$ [3] $p = 0.89$ [4] | 88 | 0.83 (0.59; 1.16), $p = 0.27$ [3] $p = 0.79$ [4] |
| Emotional Stability 3rd tercile [2] | 77 | 1.14 (0.80; 1.62), $p = 0.48$ [3] $p = 0.80$ [4] | 95 | 1.12 (0.80; 1.57), $p = 0.51$ [3] $p = 0.80$ [4] | 88 | 0.80 (0.57; 1.12), $p = 0.19$ [3] $p = 0.71$ [4] |
| Openness to Experiences 3rd tercile [1] | 87 | 1.13 (0.80; 1.61), $p = 0.48$ [3] $p = 0.89$ [4] | 95 | 0.82 (0.59; 1.14), $p = 0.23$ [3] $p = 0.79$ [4] | 109 | 1.09 (0.79; 1.52), $p = 0.60$ [3] $p = 0.89$ [4] |
| Openness to Experiences 3rd tercile [2] | 87 | 1.12 (0.79; 1.59), $p = 0.53$ [3] $p = 0.80$ [4] | 95 | 0.82 (0.59; 1.14), $p = 0.23$ [3] $p = 0.71$ [4] | 109 | 1.11 (0.80; 1.54), $p = 0.54$ [3] $p = 0.80$ [4] |
| | High Adherence to EE | | Moderate Adherence to EE | | Low Adherence to EE | |
| Extraversion 3rd tercile [1] | 91 | 1.79 (1.27; 2.53), $p < 0.001$ *,[3] $p < 0.01$ *,[4] | 51 | 1.00 (0.81; 1.23), $p = 0.98$ [3] $p = 0.98$ [4] | 84 | 0.59 (0.42; 0.83), $p < 0.01$ *,[3] $p < 0.01$ *,[4] |
| Extraversion 3rd tercile [2] | 91 | 1.76 (1.24; 2.49), $p < 0.01$ *,[3] $p < 0.01$ *,[4] | 51 | 0.98 (0.66; 1.46), $p = 0.93$ [3] $p = 0.93$ [4] | 84 | 0.61 (0.43; 0.85), $p < 0.01$ *,[3] $p < 0.01$ *,[4] |

**Table 6.** *Cont.*

| | High Adherence to UE | | Moderate Adherence to UE | | Low Adherence to UE | |
|---|---|---|---|---|---|---|
| | n | OR (CI 95%), *p* | n | OR (CI 95%), *p* | n | OR (CI 95%), *p* |
| Agreeableness 3rd tercile [1] | 123 | 1.66 (1.18; 2.34), *p* < 0.01 *,[3] *p* < 0.01 *,[4] | 82 | 1.31 (0.89; 1.91), *p* = 0.17 [3] *p* = 0.25 [4] | 126 | 0.53 (0.39; 0.73), *p* < 0.001 *,[3] *p* < 0.001 *,[4] |
| Agreeableness 3rd tercile [2] | 123 | 1.61 (1.13; 2.27), *p* < 0.01 *,[3] *p* < 0.05 *,[4] | 82 | 1.28 (0.87; 1.88), *p* = 0.21 [3] *p* = 0.31 [4] | 126 | 0.55 (0.40; 0.76), *p* < 0.001 *,[3] *p* < 0.01 *,[4] |
| Conscientiousness 3rd tercile [1] | 133 | 2.85 (2.01; 4.06), *p* < 0.001 *,[3] *p* < 0.001 *,[4] | 72 | 1.10 (0.76; 1.60), *p* = 0.62 [3] *p* = 0.65 [4] | 102 | 0.37 (0.27; 0.52), *p* < 0.001 *,[3] *p* < 0.001 *,[4] |
| Conscientiousness 3rd tercile [2] | 133 | 2.75 (1.92; 3.93), *p* < 0.001 *,[3] *p* < 0.001 *,[4] | 72 | 1.06 (0.72; 1.55), *p* = 0.76 [3] *p* = 0.81 [4] | 102 | 0.39 (0.28; 0.55), *p* < 0.001 *,[3] *p* < 0.001 *,[4] |
| Emotional Stability 3rd tercile [1] | 104 | 1.86 (1.32; 2.62), *p* < 0.001 *,[3] *p* < 0.01 *,[4] | 65 | 1.26 (0.87; 1.84), *p* = 0.23 [3] *p* = 0.31 [4] | 91 | 0.48 (0.35; 0.67), *p* < 0.001 *,[3] *p* < 0.001 *,[4] |
| Emotional Stability 3rd tercile [2] | 104 | 1.90 (1.34; 2.69), *p* < 0.001 *,[3] *p* < 0.01 *,[4] | 65 | 1.26 (0.86; 1.85), *p* = 0.24 [3] *p* = 0.32 [4] | 91 | 0.47 (0.33; 0.66), *p* < 0.001*,[3] *p* < 0.001 *,[4] |
| Emotional Stability 1st tercile (Neuroticism) [1] | 68 | 0.62 (0.44; 0.88), *p* < 0.01 *,[3] *p* < 0.05 *,[4] | 54 | 0.84 (0.57; 1.23), *p* = 0.36 [3] *p* = 0.46 [4] | 138 | 1.71 (1.24; 2.36), *p* < 0.01 *,[3] *p* < 0.01 *,[4] |
| Emotional Stability 1st tercile (Neuroticism) [2] | 68 | 0.61 (0.42; 0.86), *p* < 0.01 *,[3] *p* < 0.05 *,[4] | 54 | 0.83 (0.56; 1.22), *p* = 0.35 [3] *p* = 0.45 [4] | 138 | 1.77 (1.28; 2.47), *p* < 0.001 *,[3] *p* < 0.01 *,[4] |
| Openness to Experiences 3rd tercile [1] | 84 | 0.76 (0.54; 1.07), *p* = 0.11 [3] *p* = 0.18 [4] | 70 | 1.17 (0.80; 1.70), *p* = 0.42 [3] *p* = 0.47 [4] | 137 | 1.14 (0.83; 1.56), *p* = 0.42 [3] *p* = 0.47 [4] |
| Openness to Experiences 3rd tercile [2] | 84 | 0.75 (0.54; 1.06), *p* = 0.11 [3] *p* = 0.18 [4] | 70 | 1.17 (0.80; 1.71), *p* = 0.41 [3] *p* = 0.46 [4] | 137 | 1.14 (0.83; 1.58), *p* = 0.41 [3] *p* = 0.46 [4] |

The *p* values below the statistical significance threshold are marked with an asterisk (*) (*p* < 0.05). [1]—Logistic regression results before adjustment. [2]—Logistic regression results adjusted by BMI and age of the study participants. [3]—The value of "*p*" before the Benjamini–Hochberg correction. [4]—The value of "*p*" after the Benjamini–Hochberg correction.

Intensified extraversion increased the probability of a high adherence to emotional eating (EE) by 76%, while agreeableness increased it by 61%. Conscientiousness and emotional stability successively increased the chance of a high adherence to the emotional eating behaviour by almost 3 times and 2 times, respectively. Moreover, increased neuroticism reduced the likelihood of a high adherence to emotional eating (EE) by 39%.

## 4. Discussion

Our study focused on revealing the relationships among sugar-related dietary patterns, personality traits, and cognitive–behavioural and emotional functioning in working-age women. Many findings from our research were noteworthy. We extracted three dietary patterns, one of which was associated with an increased tendency to intake sugar and engage in less-healthy dietary behaviours, and at the same time, it was related to certain personality types. In many ways, dietary choices, as human behaviours, represent an evolutionary puzzle. We sought to determine whether selected dietary patterns related to specific eating habits might be determined by personality or other psychological traits. Generally speaking, personality traits are good correlators of dietary behaviours. Our study found associations between selected personality traits and specific eating behaviour patterns.

We discovered that conscientiousness stood out in the analyses. This trend is also continuing in Poland. Since 2010, annual sugar consumption has increased from 39.9 kg to nearly 42 kg. We revealed that working-age women with high conscientiousness had a 34% lower chance for adhering to the SWDP while 80% higher to the PHDP. It is worth noting that women with a high adherence to the SWDP were characterised by a high consumption of fried products and generally achieved a high intensity in the consumption of non-healthy dietary products (nHDI). Women with an adherence to the PHDP were characterised by high consumptions of vegetables and fruits and reached a high intensity in the consumption of healthy foods (pHDI).

We believe that highly conscientious working-age women pay more attention for selecting products in their daily diet. Perhaps when following a diet—self-imposed or imposed by a specialist—they were more eager and found it easier to follow the recommendations. Likewise, some papers report that personality can influence dietary choices [12,13]. Some authors have even shown that higher conscientiousness may reduce health risk behaviours, which would align with our results [50]. Further research into conscientiousness is required [51].

Interesting correlations were discovered between a high adherence to pro-healthy dietary patterns (PHDPs) and high intensities in uncontrolled eating (UE) and cognitive restraint (CR). Working-age women with high uncontrolled eating (UE) were 66% more likely to adhere to PHDPs, while high-CR women were 47% more likely to adhere to this DP. We hypothesise that respondents who maintain a good-quality diet have more restrictive personalities. Similarly, Jeżewska-Zychowicz et al. showed that higher levels of food involvement are associated with healthier dietary behaviours [52]. Declared restrictions on consuming foods high in sugar, fat, and starch were observed in girls in the "fruits and vegetables" dietary patterns by Galinski et al. [53].

Our study found that women with a high cognitive restraint (CR) were almost 2-fold more likely to adhere to the SWDP. In contrast, the same group was 60% less likely to adhere to the PHDP. The inconsistency of this result requires a further explanation. Cognitive restraint is the control over the food intake, which influences body weight and body shape and exerts quantitative and qualitative influences on the dietary intake [44]. Dieting for weight control is closely associated with cognitive restraint [54]. We could expect CR women to apply the principles of nutrition correctly. Nothing could be further from the truth in the case of our study. The women in our group likely lacked the necessary knowledge to maintain, for example, a healthy body weight, restricted healthy products, and consumed more sweets in the SWDP. As some authors suggest, for cognitive restraint, there is no evidence indicating whether subjects take aspects of diet quality into account and, therefore, may have a greater intake of sweet foods [55]. This hypothesis is reinforced by the fact that subjects restricting food intake can result in the adoption of unhealthy dietary habits and the potential development of eating disorders [52].

We also found that working-age women with high emotional eating (EE) were 37% less likely to follow the SWDP. Usually, sweet consumption is considered to result from succumbing to emotions. The question then arose about why women with a high EE were less likely to practise the SWDP. Research indicates that food choices depend on the emotions accompanying them. Knowing the range of emotions accompanying the consumption would be necessary to make inferences about emotional food choices. We know, for example, that individuals selected more sweets and fewer non-sweet foods when primed to feel grateful rather than proud, a positive emotion experienced by attributing a positive outcome to the self [56]. Consumption in subjects with high EE is generally associated with the intake of hyper-palatable energy-dense foods [57]. Making a judgement on the type of food consumed would require a diagnosis of the causes of the emotional state.

Conscientiousness proved to be a crucial trait when analysing trait severity versus adherence to eating behaviours—uncontrolled eating (UE), cognitive restraint (CR), and emotional eating (EE). Respondents characterised by high conscientiousness were 74%

more likely to have a high adherence to uncontrolled eating (UE), 36% less likely to have a high adherence to cognitive restraint (CR), and almost 3 times more likely to have a high adherence to emotional eating (EE). Conscientious people are portrayed as being highly organised and self-motivated and knowing what they want [31]. They stick persistently to the rules they set [31]. One supposes this accounts for the significant correlation with cognitive restraint (CR). Women knowing that restricting food is unhealthy for them are less likely to exhibit this behaviour. However, it would be necessary to test women's dietary knowledge in further studies to confirm this conjecture. Women may want to maintain their consumption of various foods because it is some reward or compensation for the day's hardships. Thus, we can assume that the high correlation with emotional eating (EE) is due to the emotional escape of women characterised by high conscientiousness. The respondents do not limit their food intake. Moreover, they consume food while being influenced by negative emotions they feel. Most likely, they cannot control the amount of food they consume—hence, the significant correlation with uncontrolled eating (UE).

In our study, we obtained another significant correlation—intensified extraversion increased the chance of a high adherence to emotional eating (EE) by as much as 76%. It can be presumed that extroverted individuals, characterised by activity, friendliness, talkativeness, and sociability, are more sensitive to stimuli received from the environment [31]. These individuals seek stimuli and pacing; they experience positive emotions [31]. However, what about when negative stimuli are more abundant and because of their sensitivity, these individuals succumb to them? The result may be the correlation we obtained. It is possible that people with intense extroversion treated food as a "springboard" and rewarded themselves with food for the accumulation of negative stimuli and emotions.

Individuals marked by agreeableness are characterised by modesty, gentleness, and affection for other people [31]. The sincerity and trust they offer the world leads us to suppose they may also be characterised by high emotional sensitivity [31]. This supposition would be confirmed by the correlation obtained in this analysis. Respondents characterised by high agreeableness were 61% more likely to have a high adherence to emotional eating (EE). Perhaps they were balancing out the emotions that overwhelmed them by consuming food.

One of the obtained significant correlations could be more apparent. According to the analysis results, working-age women characterised by emotionally solid stability were almost 2 times more likely to have a high adherence to emotional eating (EE). According to the five-factor model of personalities, enhanced emotional stability signifies the ability to cope with stress and emotional adjustment, so this result is unclear and requires further analysis [31].

Available research has proven that seasons affect working-age women's diets [58–61]. It is interesting to wonder whether, in the case of this study, we would obtain different dietary patterns depending on the season in which the respondents' data would be collected, and, if so, how many differences we would find. In addition to the impact of the seasons, other factors affecting the diet should be considered. An example of such a factor could be the lockdown during COVID-19. Changes in diets before and during the lockdown were noted [62]. The diet quality examined in this study was higher during lockdowns than in the periods before [62].

Interestingly, the literature reports that one of the Big Five personality traits, neuroticism, was positively associated with depression [63]. It could, therefore, be interesting to determine whether oxidative stress has been proven to be increased in depression, especially as the literature confirms it [64,65]. The diet's antioxidant capacity plays a vital role in counteracting oxidative stress, and it has been shown not only in our studies but also in others [66–68]. It was demonstrated that antioxidant supplementation has been proven to be associated with improvements in depression and anxiety [69]. Antioxidants and the diet's antioxidant capacity are still areas of intense scientific research [70,71]. The question arises—Is it possible to discover any correlations between the antioxidant capacity of the

diet and personality traits or eating behaviours? It would be interesting to look into this topic in future studies.

Even though this study reached its aims, it had some limitations. First, we recruited a final sample of 624 working-age women—internet users—for the study. According to the inclusion criterion, this group was supposed to have women up to 64 years old, but slightly younger respondents entered the survey. We presume that older respondents use the internet less frequently and are less willing to be surveyed through it. The survey was only based on selected questionnaires, and the participants self-reported independently via the Internet. This allowed for the collection of a limited type of data. First, this is an extensive representative-sample-size survey that can provide reliable results. Second, this study needs more information about the body compositions and nutritional knowledge of the surveyed women and possible correlations regarding these parameters. Moreover, only women could participate in this study, resulting in 112 cases being excluded. Verifying women's nutritional knowledge and body compositions, which are lacking in this case, would prove to be a definite advantage of the study. A part of the group was also students who spent lockdown time between 2020 and 2021 in family homes. Owing to the understatement of the socio-demographic section question, we wonder whether students reported their hometown or the town where they are currently studying. Therefore, we decided not to include this in confounding factors.

Ultimately, understanding the factors that advance and hinder dietary restraint is critical as more consumers face the challenge for improving their health status via nutritional modifications. In addition, understanding how to encourage healthy restraint behaviours may help in macro-environmental changes to combat civilisation's diseases. Intensities in agreeableness and openness to experience did not affect the adherence to the extracted DPs. However, significant correlations were observed between the extracted DPs and the severity of the traits of extraversion, emotional stability, and conscientiousness. Significant correlations were discovered between the SWDP and PHDP and eating behaviours, UE, CR, and EE. In further research, it will be worth considering the dietary antioxidant capacity by evaluating the diet in the context of personality traits at risk for depression and checking the nutritional knowledge of female and male respondents.

## 5. Conclusions

In accordance with the obtained results, the importance of the roles of the personality and cognitive–behavioural and emotional functioning in working-age women in forming sugar-related dietary patterns becomes very significant. This study sheds new light on the necessity for considering the mentioned psychological aspects in developing effective strategies for improving dietary habits in society.

The presented research results are a part of the current interest in health psychology seeking determinants of health behavioural changes. They can serve as practical interventions for changing health behaviours, such as prevention, counselling, or interventions for those wishing to change the dietary behaviours covered by this study.

Although various factors influence dietary choices, our study shows that personality traits and eating behaviours play essential roles, which should be considered when designing effective education and intervention programs. This insight allows us to understand the deeper motivations and mechanisms that drive human dietary choices related to the intake of sugars, the over-consumption of which is one of the leading nutritional errors in the population. Based on these results, there is a need for further research to explore the psychological and neurobiological aspects that influence dietary habits.

**Supplementary Materials:** The following supporting information can be downloaded at https://www.mdpi.com/article/10.3390/app14083176/s1, Table S1: Intensity in daily consumption of selected product groups; Table S2: The food frequency intake of the study group (n = 624) described in means and medians; Table S3: Mean food frequency intake per day for high, medium, and low adherences to the sweet-Western dietary pattern (SWDP); Table S4: Mean food frequency intake per day for high, medium, and low adherences to the pro-healthy dietary pattern (PHDP);

Table S5: Mean food frequency intake per day for high, medium, and low adherences to the dairy dietary pattern (DDP); Table S6: TIPI-PL results; Table S7: TIPI-PL intensity in features; Table S8: TFEQ-13 interpreted results.

**Author Contributions:** Conceptualisation, A.G. and M.C.-M.; methodology, A.G., M.C.-M. and B.S.; formal analysis, A.G.; investigation, A.G. and M.C.-M.; resources, M.W.; data curation, A.G.; writing—original draft preparation, A.G.; writing—review and editing, A.G., M.C.-M., B.S. and M.W.; supervision, M.C.-M.; project administration, M.C.-M.; funding acquisition, M.C.-M. All authors have read and agreed to the published version of the manuscript.

**Funding:** This research received no external funding.

**Institutional Review Board Statement:** The study protocol complied with the Declaration of Helsinki for Human and Animal Rights and its later amendments. It received ethical approval from the Board of Bioethics of the University of Medical Science (120/21).

**Informed Consent Statement:** Informed consent was obtained from all the subjects involved in the study.

**Data Availability Statement:** The data supporting the conclusions of this study are included within the article and its additional files. The other datasets used and/or analysed during the current study are available from the corresponding author upon reasonable request.

**Conflicts of Interest:** The authors declare no conflicts of interest.

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
