# Peer review of "Assessment of Sugar-Related Dietary Patterns to Personality Traits and Cognitive–Behavioural and Emotional Functioning in Working-Age Women"

_applsci, doi:10.3390/app14083176_

Round 1

Reviewer 1 Report

Comments and Suggestions for Authors

1. Abstract and Introduction: written correctly.

2. Materials and Methods – The authors indicate that 624 women aged 18 and 54 were examined. However, the results in Table 1 indicate that as many as 93% of the respondents were women aged 18-26. In the remaining age groups 27-54, there is a total of 7% (4% 27-35 years, 2% 36-44 years, 1% 45-54). Therefore, according to the Reviewer, the analysis should include only women aged 18-26, and the Authors should clarify in the work's title that these are women in the so-called "new adult". It cannot be concluded that the results apply to women in such a wide age range.

3. In Table 1, the BMI units - per m2 - standardize the superscript everywhere.

4. Table 1 shows the results obtained during the research. Therefore, it should be moved from the Methods section to the Results section.

5. In the Results section, the regression model should be presented as raw and adjusted for factors that could have influenced the results. Once the age group issue is removed, it would be BMI, and it is worth including information about the Place of residence and  Housing as well. This is particularly important in such a young age group and has implications for nutrition - the surveyed were mainly students and pupils.

6. Line 120 - The frequency of food intake is shown in Table S1. – there is no further reference to the remaining Supplementary Materials in the text. And there are them from S1 to S8. These tables present the results from the KomPAN analysis, so references to them should also be placed in the Results section and not in the Methods section.

7. Please standardize the spelling: KomPAN or KOMPAN?

8. Lines 163-164 and 166-167 – repetition. Only the second fragment should be left.

9. Reviewer's biggest problem was the number of self-cited works by authors. Out of 54 references, as many as eight are self-cited, which, in the Reviewer's opinion, can be replaced by world literature. Self-citation applies to items 10, 22, 24, 25, 44, 45, 46 and 49. In the Reviewer's opinion, only items 10 and 22 are appropriately cited and refer to the significant research results and methodology used in this work. The authors should refer the obtained research results to the research of other authors, both domestic and foreign, and not only focus mainly on comparison with their own. E.g. Line 359 -365 – The authors in this fragment quote as many as 3 of their publications.

Comments on the Quality of English Language

Minor editing of English language required.

Author Response

Authors' Response to the Reviewer's Comments

Journal:             Applied Sciences

Title:                  Assessment of sugar-related dietary patterns to personality traits, cognitive-behavioural and emotional functioning in women aged 18-54

Authors:           Agnieszka Garbacz, BogusÅ‚aw Stelcer, Michalina Wielgosik, Magdalena Czlapka-Matyasik *

Dear Reviewer,

We would like to extend our sincere gratitude for your invaluable feedback and constructive criticism of our manuscript titled "Assessment of sugar-related dietary patterns to personality traits, cognitive-behavioural and emotional functioning in women aged 18-54" Your time, effort, and expertise in reviewing our work are greatly appreciated.

We have carefully considered the comments and suggestions you and other reviewers provided. We are pleased to inform you that we have addressed all the concerns raised and have made appropriate revisions to improve the quality and clarity of the manuscript. Your insightful remarks have undoubtedly contributed to enhancing our research's overall coherence and rigour. We are truly grateful for your thorough examination and thoughtful recommendations, which have undoubtedly strengthened the scholarly integrity of our work.

Please find attached detailed responses to reviewer’s comments. Thank you once again for your time, expertise, and continued support.

Reviewer 2 Report

Comments and Suggestions for Authors

Author Response

Authors' Response to the Reviewers' Comments

Journal:             Applied Sciences

Title:                  Assessment of sugar-related dietary patterns to personality traits, cognitive-behavioural and emotional functioning in women aged 18-54

Authors:           Agnieszka Garbacz, BogusÅ‚aw Stelcer, Michalina Wielgosik,

Magdalena Czlapka-Matyasik *

Dear Reviewer,

We would like to extend our sincere gratitude for your invaluable feedback and constructive criticism of our manuscript titled "Assessment of sugar-related dietary patterns to personality traits, cognitive-behavioural and emotional functioning in women aged 18-54" Your time, effort, and expertise in reviewing our work are greatly appreciated.

We have carefully considered the comments and suggestions you and other reviewers provided. We are pleased to inform you that we have addressed all the concerns raised and have made appropriate revisions to improve the quality and clarity of the manuscript. Your insightful remarks have undoubtedly contributed to enhancing our research's overall coherence and rigour. We are truly grateful for your thorough examination and thoughtful recommendations, which have undoubtedly strengthened the scholarly integrity of our work.

Please find attached detailed response to reviewer’s comment. Thank you once again for your time, expertise, and continued support.

Reviewer 3 Report

Comments and Suggestions for Authors

Thank you for the opportunity to review a topic, namely, Assessment of sugar-related dietary patterns to personality traits, cognitive-behavioural and emotional functioning in women. This study was conducted to analyze sugar-related dietary patterns and personality traits, cognitive-behavioural and emotional functioning as a variable of eating behaviour.

However the current version of paper needs some changes. My concerns are, as follows:

1. Lines 96-97: As this study is not a medical experiment, it was exempt from ethical approval from the Poznan University of the Medical Sciences Bioethics Committee according to Polish laws and GCP regulations (decision number: 261 527/20)“.

Ethical approval is required for any human research that involves human subjects, human material, human tissues, or human data (please see information: https://www.mdpi.com/journal/applsci/instructions#ethics). 

When reporting on research that involves human subjects, human material, human tissues, or human data, authors must declare that the investigations were carried out following the rules of the Declaration of Helsinki of 1975 (https://www.wma.net/what-we-do/medical-ethics/declaration-of-helsinki/), revised in 2013. According to point 23 of this declaration, an approval from the local institutional review board (IRB) or other appropriate ethics committee must be obtained before undertaking the research to confirm the study meets national and international guidelines. As a minimum, a statement including the project identification code, date of approval, and name of the ethics committee or institutional review board must be stated in Section ‘Institutional Review Board Statement’ of the article.

2. Line 99: Recruitment was done using the snowball method“. The application of the sampling snowball method should be clarified and detailed.

3. What was the design of this study?

4. Lines 111-113: This information is the result of the study but is not the methods.

5. It seems necessary for Authors to specify the validity and reliability of some questionnaires used. 

6. It seems that the study lacks logic. What were the dependent and independent variables? It is incomprehensible what regression analysis (in terms of linear or logistic) did the Authors use to analyze the study data.

7. Additionally, I suggest that the Authors should improve statistical data analysis and/or perform additional Structural Equation Modeling (SEM). Perhaps a Biostatistician could help the Authors.

8. It is also clear that it is necessary for Authors to draw up deductive hypotheses for the present study.

9. In addition, there are so many statistical analysis tests that a Bonferroni correction is needed, too. The Bonferroni correction counteracts the family-wise error rate problem by adjusting the alpha value based on the number of tests. To find your adjusted significance level, divide the significance level (α) for a single test by the number of tests (n).

10. Line 271: “Generally speaking, personality traits are good predictors of dietary behaviours.”. As the Authors carried out a single cross-sectional study in design, it is impossible to identify any cause and effect. This is just a correlation study. Hence, the term such as “predictors” should not be used in the manuscript.

11. The manuscript lacks of limitation paragraph. Please, add it.

12. The conclusions are confusing as well as the results of this study cannot be applied in practice.

Best regards

Author Response

(The authors gave the same response as above.)

Round 2

Reviewer 1 Report

Comments and Suggestions for Authors

I thank the authors for responding to the Reviewer's comments and correcting the manuscript. Apart from two suggestions (Rev. Comments 1 and 8), all suggestions were considered.

1. The Authors explain that they used the "working age" category. If so, in the opinion of the Reviewer, it is better to use this terminology also in the title of the manuscript.

Taking into account the Authors' explanations regarding the Reviewer's suggestion to exclude people aged 27-54 from the analysis - if the Authors rechecked the statistical analysis and obtained the same statistically significant values, the Reviewer still maintains that people aged 27-54 should be excluded from results section for the previously mentioned reasons to get more reliable results from the Authors' research.

2. Number of self-cited works by authors - The Reviewer does not question the importance of indicating in the methodology of the manuscript those works that refer to the development of a validated questionnaire and its use. However, there are also other places in the manuscript, e.g. in the Discussion section, two authors' works are cited in the sentence "Available research has proven that seasons affect women's diets. [48–52]." (line 380). This sentence refers to an obtained result of the research/conclusions and not the justification for using the questionnaire or the authors' analysis experience. In the Reviewer's opinion, the Discussion part should be thoroughly verified regarding the validity of the self-citations. If the Authors claim otherwise, I leave this to the Editor's decision.

The Authors' response: "We hope the Reviewer now understands this, especially as all the other reviewers appreciated our output and accepted our carefully selected quotations. Their statement was: It does not include an excessive number of self-citations." - I leave without comment.

3. The authors added some references but did not indicate which ones they were in the text.

Author Response

Authors' Response to the Reviewer's Comments

Journal:             Applied Sciences

Title:                  Assessment of sugar-related dietary patterns to personality traits, cognitive-behavioural and emotional functioning in working-age women

Authors:           Agnieszka Garbacz, BogusÅ‚aw Stelcer, Michalina Wielgosik, Magdalena Czlapka-Matyasik *

Dear Reviewer,

We sincerely thank you for your invaluable feedback and constructive criticism of our manuscript titled "Assessment of sugar-related dietary patterns to personality traits, cognitive-behavioural and emotional functioning in working-age women" Your time, effort, and expertise in reviewing our work are greatly appreciated.

We have carefully considered the comments and suggestions you and other reviewers provided. We are pleased to inform you that we have addressed all the concerns raised and have made appropriate revisions to improve the quality and clarity of the manuscript. Your insightful remarks have undoubtedly enhanced our research's overall coherence and rigour. We are truly grateful for your thorough examination and thoughtful recommendations, which have undoubtedly strengthened the scholarly integrity of our work.

Please find attached the revised version of our manuscript, including English editing via a native speaker, with detailed responses to each reviewer’s comments. Thank you once again for your time, expertise, and continued support.

Reviewer 3 Report

Comments and Suggestions for Authors

Thank you for the opportunity to re-review a topic, namely, Assessment of sugar-related dietary patterns to personality traits, cognitive-behavioural and emotional functioning in women. This study was conducted to analyze sugar-related dietary patterns and personality traits, cognitive-behavioural and emotional functioning as a variable of eating behaviour.

Major changes are needed:

1. What was the design of this study?

2. How was the representative sample calculated for the study?

3. The flowchart is missing the size (N) of the target population from which the selection was made. The portion of the population selected should be represented by the symbol “n”.

4. How many questionnaires were sent in total?

5. In what specific ways have questionnaires been disseminated?

6. What was the response rate?

7. For Authors, it seems necessary to specify the validity and reliability of some questionnaires used.

8. What were the dependent and independent variables?

9. How did the authors assessed the body mass index of study participants? Was that a direct way? What exclusion criteria for study participants have been set depending on body mass index?

10. As the Authors have conducted more than 100 statistical tests to confirm one hypothesis of this study, therefore a Bonferroni correction is needed, too. The Bonferroni correction counteracts the family-wise error rate problem by adjusting the alpha value based on the number of tests. To find your adjusted significance level, divide the significance level (α) for a single test by the number of tests (n).

11. It seems that authors should also include males in the study. Does the problematic issue under consideration concern only the female gender?

Author Response

(The authors gave the same response as above.)
